# Comparative Analysis of Cytokine Profiles in Cerebrospinal Fluid and Blood Serum in Patients with Acute and Subacute Spinal Cord Injury

**DOI:** 10.3390/biomedicines11102641

**Published:** 2023-09-26

**Authors:** Davran Sabirov, Sergei Ogurcov, Ilya Shulman, Ilyas Kabdesh, Ekaterina Garanina, Albert Sufianov, Albert Rizvanov, Yana Mukhamedshina

**Affiliations:** 1OpenLab “Gene and Cell Technologies”, Institute of Fundamental Medicine and Biology, Kazan (Volga Region) Federal University, 420008 Kazan, Russia; 2Neurosurgical Department No. 2, Republic Clinical Hospital, 420138 Kazan, Russia; 3Department of Genetics, Institute of Fundamental Medicine and Biology, Kazan (Volga Region) Federal University, 420008 Kazan, Russia; 4Department of Neurosurgery, Sechenov First Moscow State Medical University of the Ministry of Health of the Russian Federation (Sechenov University), 119991 Moscow, Russia; 5The Research and Educational Institute of Neurosurgery, Peoples’ Friendship University of Russia (RUDN), 117198 Moscow, Russia; 6Department of Histology, Cytology and Embryology, Kazan State Medical University, 420012 Kazan, Russia

**Keywords:** traumatic spinal cord injury, cytokine profile, cerebrospinal fluid, blood serum, multiplex analysis

## Abstract

Background: Cytokines are actively involved in the regulation of the inflammatory and immune responses and have crucial importance in the outcome of spinal cord injuries (SCIs). Examining more objective and representative indicators of the patient’s condition is still required to reveal the fundamental patterns of the abovementioned posttraumatic processes, including the identification of changes in the expression of cytokines. Methods: We performed a dynamic (3, 7, and 14 days post-injury (dpi)) extended multiplex analysis of cytokine profiles in both CSF and blood serum of SCI patients with baseline American Spinal Injury Association Impairment Scale grades of A. Results: The data obtained showed a large elevation of IL6 (>58 fold) in CSF and IFN-γ (>14 fold) in blood serum at 3 dpi with a downward trend as the post-traumatic period increases. The level of cytokine CCL26 was significantly elevated in both CSF and blood serum at 3 days post-SCI, while other cytokines did not show the same trend in the different biosamples. Conclusions: The dynamic changes in cytokine levels observed in our study can explore the relationships with the SCI region and injury severity, paving the way for a better understanding of the pathophysiology of SCI and potentially more targeted and personalized therapeutic interventions.

## 1. Introduction

Spinal cord injury (SCI) is a global health problem affecting tens of thousands of people every year, with profound consequences for those affected and significant societal costs. Neurological disorders resulting from SCI often lead to severe disability and dramatically change the lives of patients and their families [1]. While SCIs used to predominantly affect younger individuals, its incidence is now equally high among the elderly [2].

The pathogenesis of SCI involves a complex interplay of primary and secondary mechanisms. The primary injury is immediate, resulting from mechanical trauma, while secondary events, such as inflammation, ischemia, and excitotoxicity, further exacerbate the injury. This complexity has necessitated research into novel and often overlooked therapeutic strategies. For instance, the work of Turczyn et al. (2022) delves into the yet inconclusive but promising role of omega-3 fatty acids in SCI treatment. Similarly, the DISCUS trial led by Saadoun et al. (2023) is pioneering in its evaluation of duroplasty for cervical SCI, a technique not yet mainstream but showing potential for significant impact [3,4].

Concurrently, the management of SCI imposes a significant burden on global healthcare systems. The lifetime care for an individual with SCI can escalate to several million dollars, depending on the severity of the injury. This is further complicated by the long-term pharmacological management required for pain, spasticity, and secondary complications. The pharmacoeconomic impact is substantial, as evidenced by ongoing trials and studies, including those investigating the protective effects of hydrogen gas against spinal cord ischemia-reperfusion injury [5].

Despite significant advances in medical, surgical, and rehabilitation care for people with SCI, measures and interventions to ensure neurological recovery remain limited. This limited recovery is primarily due to the central nervous system’s (CNS) inherently low regenerative capacity, the complexity of the injury, and the poorly understood pathophysiological events after SCIs [6].

The complexity of SCI diagnosis and treatment is further highlighted by the potential for misdiagnosis with other spinal conditions. For instance, chronic non-bacterial osteomyelitis (CNO) is often confused with infectious spondylodiscitis or malignant lesions in the spine, complicating the diagnostic process and potentially leading to ineffective treatments [7]. This underscores the need for a comprehensive approach to spinal disorders, including SCI, to minimize treatment-related morbidity.

Evaluating new treatments for SCI presents a significant challenge. Demonstrating neurological recovery during the acute phase is particularly challenging due to the potential for spontaneous recovery [8]. This problem is exacerbated by the fact that in the clinical evaluation of new treatments, we rely on a functional neurological examination only [9]. Although tools, such as the International Standards for Neurological Classification of Spinal Cord Injury (ISNCSCI), provide a standardized assessment of neurological function, it has limitations when used in clinical trials [10]. This highlights the importance of examining other, more objective and representative indicators of the patient’s condition, such as biomarkers, for example.

In light of the need to search for more reliable and stable biomarkers in SCI, it seems more representative to study the cerebrospinal fluid (CSF), which is in close contact with the CNS and can provide a more accurate picture of the activity of various molecules in the area of injury. However, it should be noted that blood sampling is more acceptable and safer, and blood serum analysis can also provide information on some key biomarkers, although it requires additional research to clarify its value as a possible diagnostic and prognostic tool in the context of SCI.

Analysis of changes in cytokine levels at different periods after injury and their correlation with the American Spinal Injury Association (ASIA) Impairment Scale score can provide valuable insights not only into the pathophysiology of SCI but also potentially guide therapeutic interventions [11,12]. By including more samples of SCI patients and using multiplex analysis, our aim was to determine how cytokine change correlates with time after severe injury, which may help develop new strategies for early diagnosis and personalized treatment of SCI.

## 2. Materials and Methods

### 2.1. Study Population

All patients (*n* = 40) were recruited at the Neurosurgical Department No. 2 of the Republican Clinical Hospital (Kazan, Russia). Before proceeding, each participant gave their written consent to have cerebrospinal fluid (CSF) and blood serum samples collected for the study. The Local Ethical Committee of Kazan Federal University granted study approval (Protocol No. 3, dated 23 March 2017).

For this prospective study, the patient selection criteria for acute and subacute SCI were as follows: individuals older than 18; SCI locations between vertebrae C3 and L3; capability for valid neurological assessment; and initial classification as ASIA grade A. An experienced research neurologist evaluated the baseline severity of neurological damage using ASIA grade assessments. Follow-up neurological checks took place at 1- and 2-weeks post-injury to determine ASIA conversion and possible motor and sensory score improvement. Among the SCI patients, injuries were located in the cervical (C) region for 18 patients, thoracic (Th) for 14 patients, and lumbar (L) for 8 patients. Of the 40 patients, 35 (87%) were male, and the mean age was 42.2 ± 15.1 years.

For the control group, serum samples were drawn from 16 physically healthy individuals who willingly consented to venous blood collection. Uninjured subjects were recruited from the Kazan Federal University student and academic population and hospital staff. The CSF samples were collected from conditionally healthy patients who sought medical attention due to lumbar disc herniations or stenosis, from whom we obtained written informed consent. The criteria for including uninjured control participants were being older than 18; no past spinal cord or brain injuries; absence of key indicators of acute or chronic inflammation and autoimmune diseases; and a standard full blood count.

### 2.2. Samples Collection and Storage

The CSF and venous blood samples were obtained from SCI patients at 3, 7, and 14 dpi. The results of blood serum multiplex analysis at 14 dpi were presented by us earlier [13]. The CSF samples from the uninjured control group were collected the day before lumbar spine surgery. Venous blood was collected via standard venipuncture in 6 mL vacuum test tubes (Apexlab, Moscow, Russia). Following a 30-min coagulation period, the blood underwent centrifugation at a speed of 3000 RPM. Subsequently, it was partitioned into 300 µL fractions and preserved at a temperature of −80 °C until the time of analysis.

The CSF samples were collected following a strict aseptic technique; a lumbar puncture with an atraumatic needle (Medispine, 20G, Indore, Madhya Pradesh, India) was performed at L3–4, and a 3 mL sample of CSF was collected. The CSF samples were divided into 500 µL aliquots, centrifuged at 1000 RPM for 10 min, and the supernatant was then immediately frozen and stored at −80 °C. All obtained samples from injured patients and the uninjured control group underwent identical procedures and were stored for comparable durations.

It is important to note that some patients with SCI may develop cerebrospinal fluid dynamics disorders, which can make it difficult to collect cerebrospinal fluid at various stages after injury. Because of this, we were unable to obtain data for all 40 patients at investigated time points after injury. This means that the number of samples used in the analysis may vary depending on the time of collection.

### 2.3. Multiplex Analysis

Within the scope of our research, we assessed fluctuations in the cytokine composition of cerebrospinal fluid (CSF) and blood serum from patients with spinal cord injuries (SCI) at different dpi. We employed xMAP Luminex technology for multiplex analysis. The Bio-Plex Pro™ #171AK99MR2 (Bio-Rad, Hercules, CA, USA) facilitated the simultaneous examination of a panel of 40 human cytokines (CCL21, CXCL13, CCL27, CXCL5, CCL11, CCL24, CCL26, CX3CL1, CXCL6, GMCSF, CXCL1, CXCL2, CCL1, IFN-γ, IL1b, IL2, IL4, IL6, CXCL8/IL8, IL10, IL16, CXCL10, CXCL11, MCP-1, MCP-2/CCL8, MCP-3, MCP-4, CCL22, MIF, MIG/CXCL9, MIP-1a/CCL3, MIP-1b/CCL4, MIP-3a/CCL20, MIP-3b/CCL19, MPIF-1/CCL23, CXCL16, SDF-1/CXCL12, CCL17, CCL25 and TNFα), using just 50 µL of the sample. All serum specimens were collectively analyzed in a single assay setup. All the CSF and blood serum samples were analyzed simultaneously within a single assay.

### 2.4. Statistical Analysis

Data analysis was conducted using version 3.6.3 of R (R Foundation for Statistical Computing, Vienna, Austria). Quantitative variables are illustrated through their mean and standard deviation, as well as their median and interquartile ranges. Before choosing the appropriate statistical tests, we utilized the Pearson test to evaluate data normality. The Kruskal–Wallis test was used to check for an overall difference in median levels of cytokines between groups, while the Dunn’s correction was used for multiple comparisons. We used the Benjamini–Hochberg method for adjustments related to multiple comparisons.

## 3. Results

### 3.1. Dynamics of Cerebrospinal Fluid Cytokine Profile after Spinal Cord Injury

We analyzed 40 cytokines in the CSF of 35 patients at various days post-SCI, revealing significant changes in cytokine concentrations at 3, 7, and 14 dpi (Figure 1 and Figure 2). In particular, we found a significant increase in the levels for CCL22 (2.7-fold, P.adj < 0.009), CCL26 (6.6-fold, P.adj < 0.017), IL8 (6.7-fold, P.adj < 0.013), CCL23 (8.2-fold, P.adj < 0.04), and IL6 (58.8-fold, P.adj < 0.001) at 3 dpi compared to uninjured control samples (Figure 3, Appendix A). At 7 and 14 dpi we continued to observe elevated levels of the abovementioned cytokines, although their concentrations were somewhat lower than at 3 dpi.

At 14 dpi, expression levels of CCL22 (1.3-fold), IL8 (2-fold), IL6 (2.6-fold), CCL23 (2.8-fold), and CCL26 (4.2-fold) were also increased compared to the uninjured control samples. However, it is important to note that the abovementioned changes were only statistically significant for CCL26 [SCI 8.69 (1.83–18.32) vs. the uninjured control 2.06 (0.44–3.06), P.adj < 0.012]. In addition, we found a significant difference in IL6 levels between the 3 and 7 dpi groups with a downward trend as the post-traumatic period increases.

We segregated cytokine levels based on the region of injury for determination of whether the region of SCI affects CSF cytokine levels. There were no discernible differences in cytokine levels among patients with varying SCI regions and the uninjured controls at 3 dpi. However, we observed that CCL17 levels in patients with C injury were significantly elevated compared to SCI at the L region and uninjured control subjects at 7 dpi, showing increases of 45-fold (P.adj < 0.005) and 10-fold (P.adj < 0.045), respectively (Figure 4A). SCI at the Th region had CCL3 levels in two-fold (P.adj < 0.05) and four-fold (P.adj < 0.003) higher compared to SCI at the C region and uninjured control subjects at 14 dpi, accordingly (Figure 4B).

### 3.2. Dynamics of Blood Serum Cytokine Profile after Spinal Cord Injury

Based on the analysis of blood serum, we found significant changes in cytokine profiles after SCI. Out of the 40 cytokines we analyzed at 3 and 7 dpi, the levels of CCL26, CXCL6, GMCSF, IFN-γ, IL1b, IL4, IL8, IL10, IL16, CXCL11, CCL8, CCL7, CXCL9, CCL3, CCL19, CCL17 showed significant deviations when compared to uninjured control samples (Figure 5, Appendix A).

We found a significant increase in the levels for CXCL6 (2.8-fold, P.adj < 0.008), IL4 (4.5-fold, P.adj < 0.04), CCL7 (5.7-fold, P.adj < 0.01), CCL26 (6-fold, P.adj < 0.002), and IFN-γ (26.9-fold, P.adj < 0.0001) at 3 dpi compared to uninjured control samples (Figure 6). At 7 dpi, the levels of the abovementioned cytokines also remained elevated: CXCL6 (4,2-fold, P.adj < 0.0002), IL4 (4,8-fold, P.adj < 0.0004), CCL7 (1.7-fold, P.adj < 0.014), CCL26 (5,7-fold, P.adj <0.01), and IFN-γ (14,9-fold, P.adj < 0.0005).

We also detected a significant decrease in several other cytokines at 3 and 7 dpi, including CCL3, CCL8, IL8, CCL19, IL10, IL16, IL1b, CXCL9, CXCL11, GMCSF, CCL17 (Figure 7). In particular, the next cytokines were lower compared to the uninjured control at 3 dpi: CCL8 (2.5-fold, P.adj < 0.05), CXCL11 (2.9-fold, P.adj < 0.005), IL8 (2.9-fold, P.adj < 0.02), IL10 (4-fold, P.adj < 0.0005), IL1b (7-fold, P.adj < 0.0001), and CXCL9 (7.4-fold, P.adj < 0.0001). At 7 dpi we continued to observe decreased levels of the abovementioned cytokines: CCL8 (2.9-fold, P.adj < 0.007), CXCL11 (8.7-fold, P.adj < 0.0008), IL8 (2.6-fold, P.adj < 0.02), IL10 (21.4-fold, P.adj < 0.0001), IL1b (10.7-fold, P.adj < 0.0001), and CXCL9 (10.7-fold, P.adj < 0.0001). In addition, CCL17 (324.5-fold, P.adj < 0.0001), GMCSF (21.2-fold, P.adj < 0.009), CCL19 (19.9-fold, P.adj < 0.003), and IL16 (4.6-fold, P.adj < 0.004) all showed significant decreases in the SCI patient samples at 7 dpi compared to the uninjured control. It is important to note that we did not observe significant differences in the blood serum cytokine profile when comparing 3 and 7 dpi.

We also segregated cytokine levels based on the region of injury for determination of whether the region of SCI affects blood serum cytokine levels. It is interesting to note that changes in cytokine levels associated with the region of injury were detected only at 3 dpi. The CCL27 level in patients with C injury was elevated 560-fold (P.adj < 0.027) and 360-fold (P.adj < 0.034) compared to SCI at the L region and uninjured control subjects, accordingly (Figure 8). The concentration of IL2 was observed to significantly increase by 2.7-fold (P.adj < 0.012) and 2.5-fold (P.adj < 0.001) in patients with C injury compared to those with SCI at the Th region and uninjured control subjects, respectively. While the concentrations of CCL17 [0.08 (0.08–4.34)], GMCSF [0.42 (0.19–4.58)], and TNFa [0.34 (0.02–0.84)] in patients with Th injury decreased compared to SCI at the C region in 566-fold, 28-fold, and 10-fold (CCL17 [45.34 (13.03–55.02)], GMCSF [11.97 (10.25–16.76)], TNFa [3.41 (2.84–5.95)]), and uninjured control subjects in 76-, 20-, and 3-folds (CCL17 [25.96 (7.71–39.98)], GMCSF [8.81 (7.62–17.18)], TNFa [1.12 (0.8–2.78)]).

## 4. Discussion

In our current study, we performed a multiplex analysis of 40 cytokines in the CSF and serum of patients with SCI at different time periods after injury (3, 7, and 14 dpi), comparing the obtained data with those of uninjured control subjects. We observed both upregulation and downregulation of various cytokines after SCI.

One of the most notable results was a sustained increase in CCL26 levels in SCI in both CSF by 3 and 14 dpi and in serum by 3 and 7 dpi. These results align with our previous study, suggesting a potentially universal role for CCL26 in the inflammatory response following traumatic SCI, regardless of the severity of the injury [13]. Elevated levels of CCL26, both in CSF and blood serum, have been observed in numerous diseases, including Alzheimer’s disease [14], asthma [15], rheumatoid arthritis [16], cancer [17], and experimental autoimmune encephalomyelitis [18]. This suggests that CCL26 alone might not provide a definitive diagnostic tool due to its broad involvement in inflammatory responses; it could potentially be used in conjunction with other specific biomarkers to enhance the accuracy of early diagnosis and to tailor personalized treatment strategies for SCI patients.

In our study, we observed a significant increase in IL8 level in the CSF of patients following SCI, while a decrease in IL8 level was noted in the blood serum of the same patient group at 3 dpi. This contrasting pattern suggests a differential regulation of IL8 in the CNS and the systemic circulation following SCI, highlighting the complex and dynamic nature of the inflammatory and immune response to such injuries. Similar results on the level of expression of IL8 in CSF and blood serum were observed by Kossmann et al. (1997) examining patients with TBI. It has been suggested that IL8 release in the CSF after neurotrauma may be associated with BBB dysfunction and nerve growth factor production [19].

Elevated levels of IL6 were also found in the CSF of SCI patients at 3 dpi. The increase in the abovementioned cytokine is consistent with the results of a study by Kwon et al. (2010) where multiplex analysis found that IL6 levels in the CSF of SCI patients are extremely high, especially in patients with severe damage [20]. These results highlight the important role of IL6 and IL8 in the acute inflammatory response after SCI. IL6, for example, is significantly increased within hours of injury in both rodents and humans, and blocking its receptor (IL6R) has been shown to reduce glial scar, neutrophil and macrophage invasion, and improve functional recovery after SCI [21,22]. Using electrochemiluminescence, Fertleman et al. (2022) found that levels of many cytokines in CSF, including IL6 and IL8, are significantly increased after orthopedic surgery. The next day after the operation, their concentration even exceeds the figures recorded immediately after surgery [23]. Based on the above data, it can be assumed that after spinal decompression surgery for SCI, the levels of these cytokines may also increase. Apparently, IL6 and IL8 alone might not provide a definitive diagnostic tool but can be used in combined biomarker approaches.

In our study, we also observed a significant increase in the CCL22 level in the CSF of patients with SCI at 3 dpi. Functionally, CCL22 is a potent chemoattractant for antigen-testing but non-resting T-lymphocytes, as well as for NK cells and monocytes. CCL22 levels in CSF have been reported to be elevated in women with multiple sclerosis [24]. These data, combined with our observations, highlight the potential importance of CCL22 as an inflammatory marker not only in neurodegenerative diseases, such as multiple sclerosis, but also in acute SCI.

We observed a sharp decrease in the pro-inflammatory cytokine IL1b at 3 and 7 dpi in blood serum, mirroring trends in our previous SCI study in rats [25]. In the study performed by Biglari et al., IL1b concentration fluctuated greatly between 4 h and 7 dpi. However, between 1 and 4 weeks after injury, the level of IL1b in blood serum significantly decreased only in those patients in whom the improvement was less significant [11].

A significant decrease in blood serum chemokines CXCL9 and CXCL11 was found by our investigation after SCI at 3 and 7 dpi. Three structurally related chemokines (CXCL9, CXCL10, CXCL11) form the subgroup, the interferon (IFN)-inducible non-ELR CXC chemokine [26]. The chemokines CXCL9 and CXCL11 function as central modulators of the immune response, predominantly in inflammatory cascades. Stein et al. (2013) in a pilot study on the simultaneous assessment of the level of 21 cytokines in the blood plasma of patients with chronic (>1 year from initial injury) SCI, on the contrary, found a significant increase in CXCL9 levels in patients with chronic SCI compared with uninjured control subjects [27].

We also found that the level of IFN-γ in the blood serum of patients was significantly increased by both 3 and 7 dpi, which may indicate inflammation and secondary damage that occurs after SCI [28]. IFN-γ has been previously shown to be involved in antimicrobial immunity and is elevated in various bacterial diseases [29]. This cytokine is also known for its role in antitumor immunity [30]. Monitoring of this cytokine can provide insight into the severity of injury and the body’s response. However, since the level of IFN-γ is also increased in other pathological conditions, its use will only be justified along with other specific biomarkers.

Our data confirm that the inflammatory and immune response to SCI is a complex and multifaceted process involving various molecules. Unfortunately, the assessment of cytokine or chemokine levels separately cannot be a valuable tool for diagnosing and monitoring the status of patients with SCI. That is why the combined biomarker approach could offer a more comprehensive understanding of the individual patient’s inflammatory profile, thereby improving the precision of therapeutic interventions. Despite the increased activity of researchers in this area, additional investigations are needed to better understand the role of injury-responsive cytokines in the context of SCI and their potential application in clinical practice.

## 5. Study Limitations

Although our study provides valuable insights, it is not without its shortcomings. This study was limited to just 40 patients with SCI, requiring larger sample sizes in future studies. The limited number of participants likely reduced the study’s ability to detect significant cytokine patterns. We encountered challenges in simultaneously collecting blood and cerebrospinal fluid from all SCI patients, attributed to cerebrospinal fluid dynamics disorders arising in certain posttraumatic phases. It should also be noted that there is a need to confirm the results using other methods of molecular analysis.

## 6. Conclusions

In light of the need to find more reliable and stable SCI biomarkers, we conducted this study, which greatly expands on our previous analysis. In an earlier published paper, we identified several cytokines in blood serum that could potentially serve as biomarkers for the severity of SCI [13]. However, given the active involvement of cytokines in the regulation of the inflammatory and immune response and their possible crucial importance for the outcome of SCI [28], we hypothesized that analysis of blood serum alone may not fully reflect the actual activity of cytokines at the site of injury in the CNS. In this context, in the current study, we extended our analysis to cytokine profiles in both CSF and blood serum in more patients and at various stages after SCI. This allowed us to explore in more detail the dynamics of change and the relationship between cytokine levels, injury severity, and recovery time, paving the way for a better understanding of the pathophysiology of SCI and potentially more targeted and personalized therapeutic interventions.

## Figures and Tables

**Figure 1 biomedicines-11-02641-f001:**
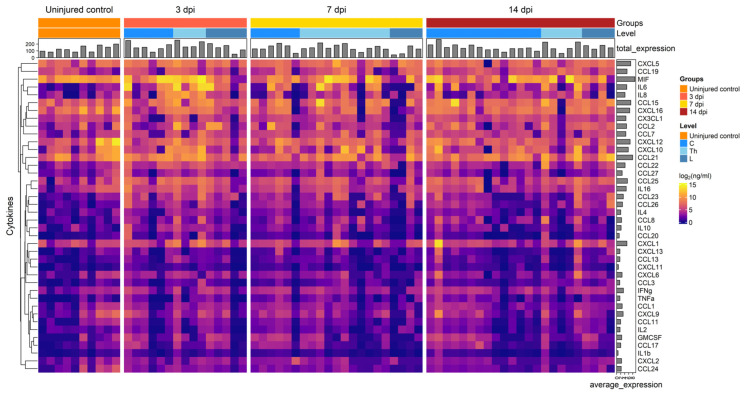
Graphical representation showing log2 cytokine concentrations (color keys), generated with the multiplex analysis of the CSF collected at 3 dpi (*n* = 15), 7 dpi (*n* = 21), 14 dpi (*n* = 23), or from uninjured control (*n* = 10). A dendrogram resulting from hierarchical clustering of cytokines is shown on the left.

**Figure 2 biomedicines-11-02641-f002:**
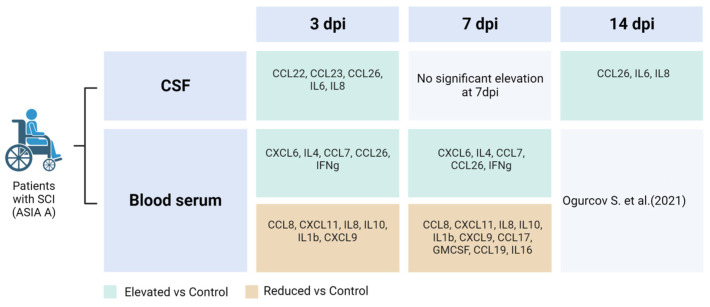
List of cytokines with statistically significant elevation or reduction in CSF and blood serum at 3 dpi, 7 dpi, and 14 dpi [13], compared to uninjured control.

**Figure 3 biomedicines-11-02641-f003:**
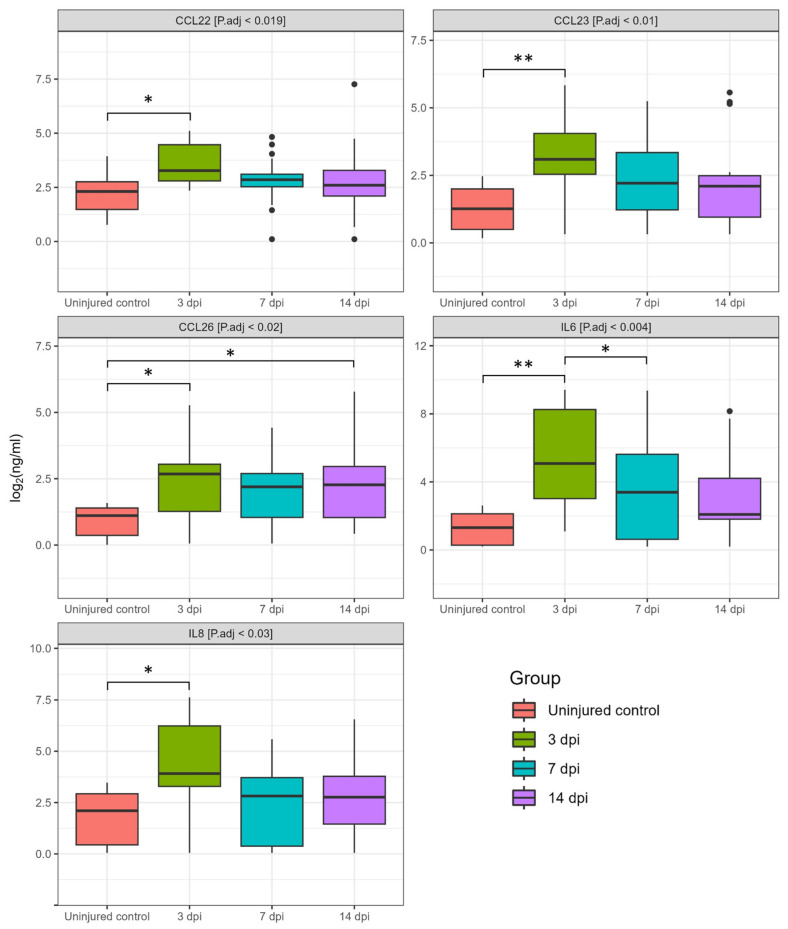
Log2-transformed CSF cytokine concentrations (ng/mL) between 3 dpi (green columns, *n* = 15), 7 dpi (blue columns, *n* = 21), 14 dpi (purple columns, *n* = 23) patients and uninjured subjects (red columns, *n* = 10). * P.adj < 0.05, ** P.adj < 0.01.

**Figure 4 biomedicines-11-02641-f004:**
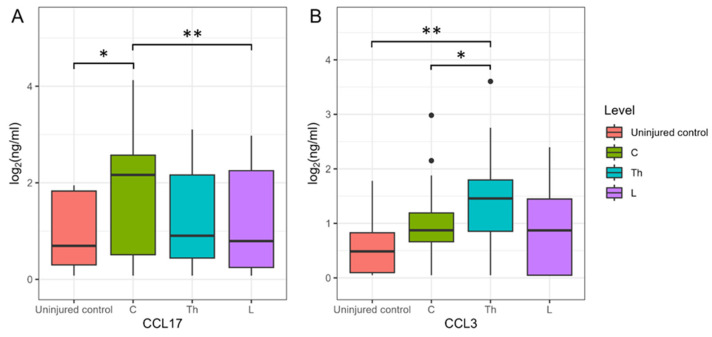
Log2-transformed CSF CCL17 and CCL3 concentrations (ng/mL) at 7 (**A**) and 14 (**B**) dpi, considering the cohort of cervical (C, *n* = 14), thoracic (Th, *n* = 11), and lumbar (L, *n* = 4) patients and uninjured subjects (red columns, *n* = 10). * P.adj < 0.05, ** P.adj < 0.01.

**Figure 5 biomedicines-11-02641-f005:**
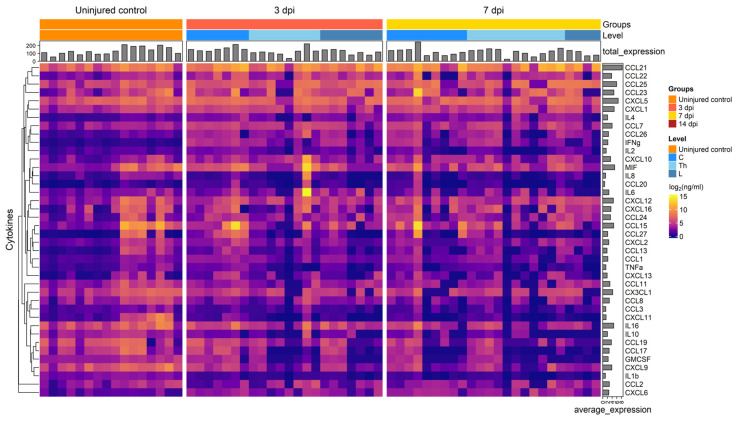
Graphical representation showing log2 cytokine concentrations (color keys), generated with the multiplex analysis of the blood serum collected at 3 dpi (*n* = 22), 7 dpi (*n* = 24), or from uninjured control (*n* = 16). A dendrogram resulting from hierarchical clustering of cytokines is shown on the left.

**Figure 6 biomedicines-11-02641-f006:**
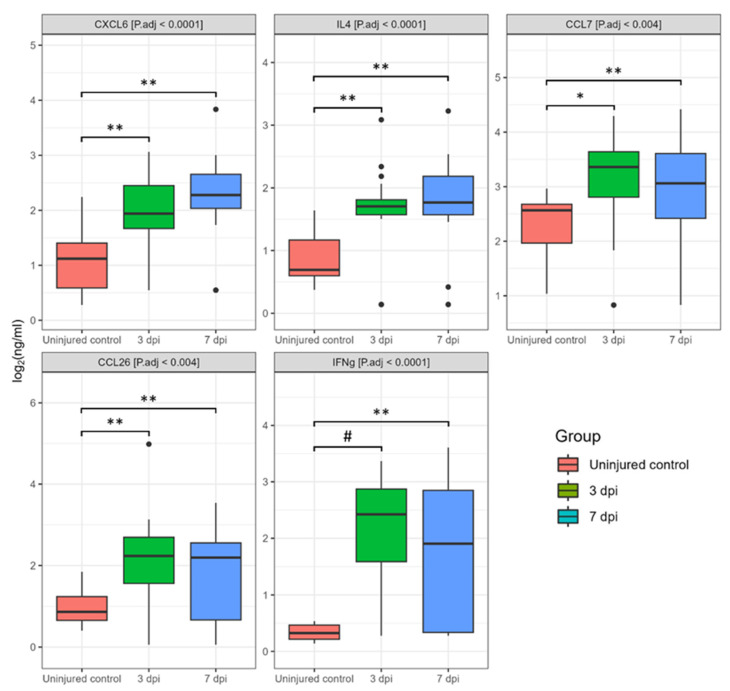
Log2-transformed blood serum cytokine concentrations (ng/mL) between 3 dpi (green columns, *n* = 22), 7 dpi (blue columns, *n* = 24) patients, and uninjured subjects (red columns, *n* = 16). * P.adj < 0.05, ** P.adj < 0.01, # P.adj < 0.0001.

**Figure 7 biomedicines-11-02641-f007:**
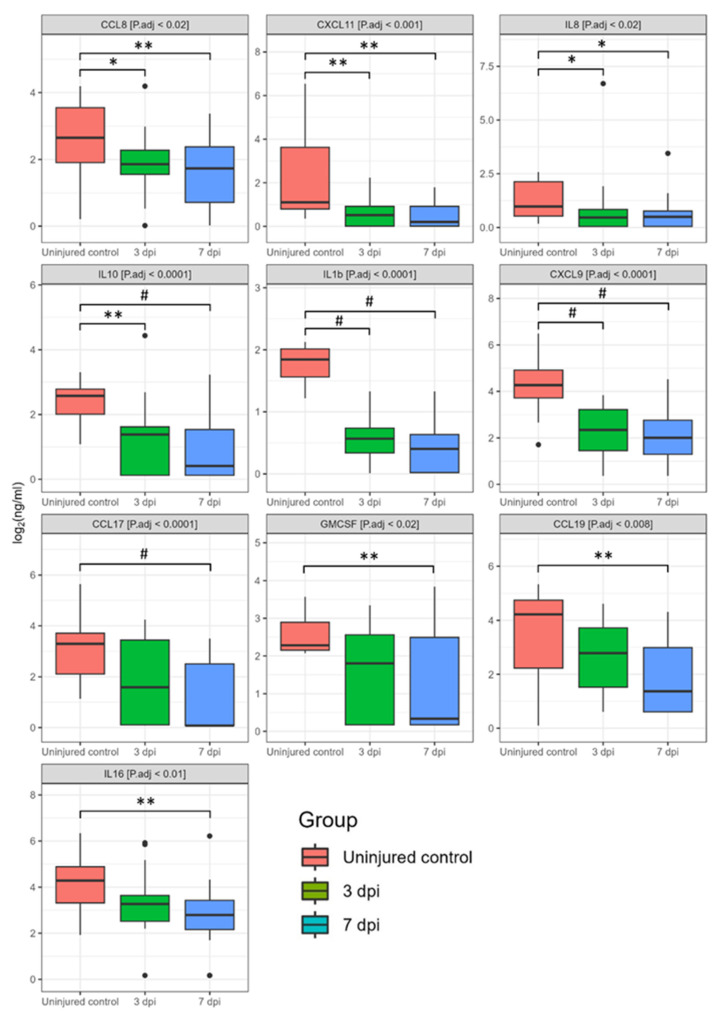
Log2-transformed blood serum cytokine concentrations (ng/mL) between 3 dpi (green columns, *n* = 22), 7 dpi (blue columns, *n* = 24) patients and uninjured subjects (red columns, *n* = 16). * P.adj < 0.05, ** P.adj < 0.01, # P.adj < 0.0001.

**Figure 8 biomedicines-11-02641-f008:**
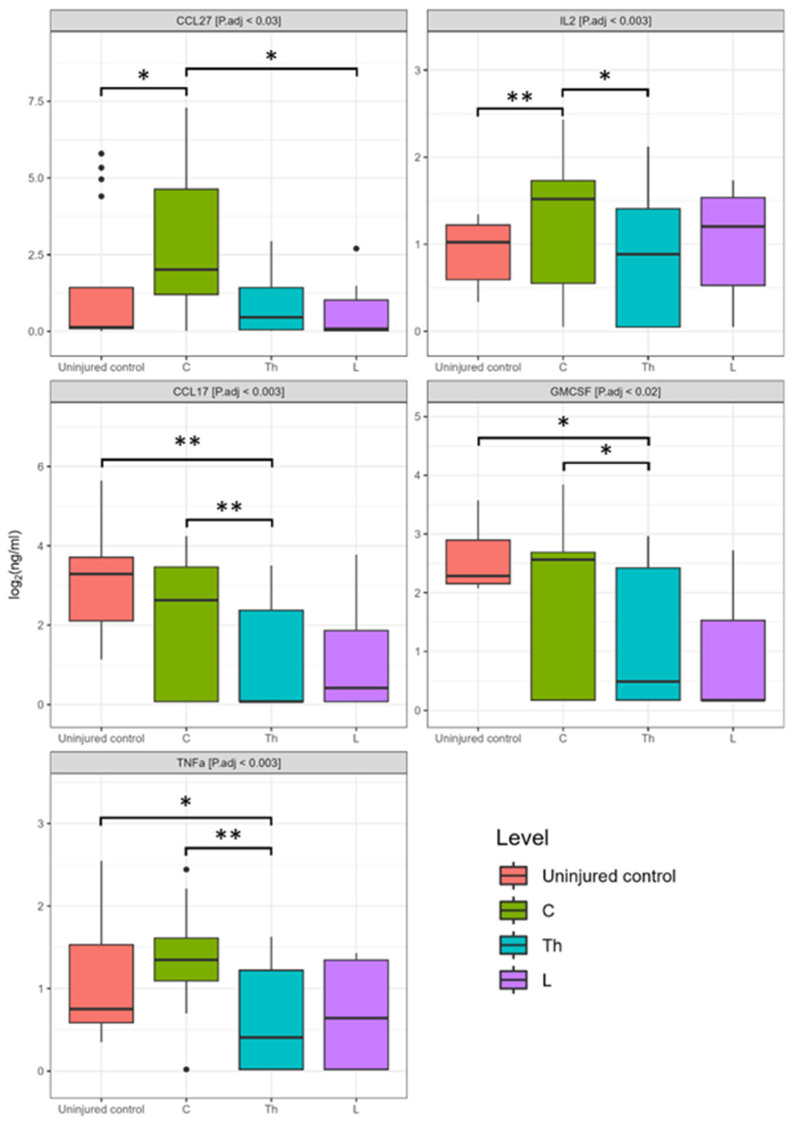
Log2-transformed blood serum cytokine concentrations (ng/mL) between patients at 3 dpi injury, considering the cohort of cervical (C, *n* = 7), thoracic (Th, *n* = 8), and lumbar (L, *n* = 7) patients, and uninjured control subjects (*n* = 16). * P.adj < 0.05, ** P.adj < 0.01.

## Data Availability

The data presented in this study are available on request from the corresponding author. The data are not publicly available due to the evolving nature of the project.

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
