# Peer review of "Comparative Analysis of Cytokine Profiles in Cerebrospinal Fluid and Blood Serum in Patients with Acute and Subacute Spinal Cord Injury"

_biomedicines, 2023, doi:10.3390/biomedicines11102641_

Round 1
Reviewer 1 Report
Manuscript presentes interesting results related to the analysis of the cytokine profiles in the CSF and Blood Serum in patients with SCI.
In the abstract, the dpi has to be defined before the first mentioning.
The figures are of low quality, and it is hard to read the marks related to the groups.
I understand that results have a lot of variables, but it is quite difficult for reader to follow all the connections between the different varaibles.
Maybe the results should be presented only in form of words, and nto numbers, as there is a supplementary table.
Author Response
Reviewer #1
Comments and Suggestions for Authors
Manuscript presentes interesting results related to the analysis of the cytokine profiles in the CSF and Blood Serum in patients with SCI.
In the abstract, the dpi has to be defined before the first mentioning.
Authors: Thank you for pointing this out. We have now defined "dpi" (days post-injury) before its first mention in the abstract to ensure clarity for the reader.
The figures are of low quality, and it is hard to read the marks related to the groups.
Authors: We have revised them to improve their resolution and readability.
I understand that results have a lot of variables, but it is quite difficult for reader to follow all the connections between the different varaibles.
Authors: We appreciate your feedback on the complexity of the results section. To address this concern, we have included a comprehensive illustration that visually represents the cytokines that showed significant changes. This can help guide the reader through the complex interplay of variables in our study.
Maybe the results should be presented only in form of words, and nto numbers, as there is a supplementary table.
Authors: Thank you for the suggestion. To clarify, we have included a graphical representation as Figure 2 in the main manuscript to make the results more accessible and easier to understand. We believe that this approach, along with the narrative in the text, provides a balanced and comprehensive view of our findings.
We have made all the suggested changes in the manuscript, which can be easily tracked in the Word document using the 'Track Changes' feature. We appreciate your valuable feedback and hope that these revisions meet your approval.

Reviewer 2 Report
The article by Sabirov et al "Comparative Analysis of Cytokine Profiles in Cerebrospinal Fluid and Blood Serum in Patients with Acute and Subacute Spinal Cord Injury" covers a potentially interesting and emerging topic
related to the diagnostics and therapy of spinal cord injury (SCI). In this sense, this remains to be potentially interesting for the Biomedicines readers.
I regard the main point of this paper as highly attractive as well as the results are clearly presented. The text does not contain any major errors, therefore I
have some minor comments and recommendations:
1. There is a need to provide slightly more expanded introduction
mentioning/describing pathogenesis of SCI and its impact of modern
healthcare/pharmacoeconomics
2. The figure summarizing and clarifying the results should be added.
3. Following references should be added and properly cited within the main text:
- Turczyn P, Wojdasiewicz P, Poniatowski ŁA, Purrahman D, Maślińska M, Żurek G, Romanowska-Próchnicka K, Żuk B, Kwiatkowska B, Piechowski-Jóźwiak B, Szukiewicz D. Omega-3 fatty acids in the treatment of spinal cord injury: untapped potential for therapeutic intervention? Mol Biol Rep. 2022 Nov;49(11):10797-10809. doi: 10.1007/s11033-022-07762-x.
- Saadoun S, Grassner L, Belci M, Cook J, Knight R, Davies L, Asif H, Visagan R, Gallagher MJ, Thomé C, Hutchinson PJ, Zoumprouli A, Wade J, Farrar N, Papadopoulos MC. Duroplasty for injured cervical spinal cord with uncontrolled swelling: protocol of the DISCUS randomized controlled trial. Trials. 2023 Aug 7;24(1):497. doi: 10.1186/s13063-023-07454-2.
- Kubaszewski Ł, Wojdasiewicz P, Rożek M, Słowińska IE, Romanowska-Próchnicka K, Słowiński R, Poniatowski ŁA, Gasik R. Syndromes with chronic non-bacterial osteomyelitis in the spine. Reumatologia. 2015;53(6):328-36. doi: 10.5114/reum.2015.57639.
- Kimura A, Suehiro K, Mukai A, Fujimoto Y, Funao T, Yamada T, Mori T. Protective effects of hydrogen gas against spinal cord ischemia-reperfusion injury. J Thorac Cardiovasc Surg. 2022 Dec;164(6):e269-e283. doi: 10.1016/j.jtcvs.2021.04.077. Epub 2021 May 4. PMID: 34090694.
4. In some places the use of English could be improved on.
Completing this gaps will have an impact on the understanding the aim of the study and, from my point of view, is absolutely necessary.
minor changes
Author Response
Reviewer #2
Comments and Suggestions for Authors
The article by Sabirov et al "Comparative Analysis of Cytokine Profiles in Cerebrospinal Fluid and Blood Serum in Patients with Acute and Subacute Spinal Cord Injury" covers a potentially interesting and emerging topic
related to the diagnostics and therapy of spinal cord injury (SCI). In this sense, this remains to be potentially interesting for the Biomedicines readers.
I regard the main point of this paper as highly attractive as well as the results are clearly presented. The text does not contain any major errors, therefore I
have some minor comments and recommendations:
- There is a need to provide slightly more expanded introduction
mentioning/describing pathogenesis of SCI and its impact of modern
healthcare/pharmacoeconomics
Authors: Thank you for your suggestion. We have expanded the introduction to include a more detailed discussion on the pathogenesis of SCI and its impact on healthcare and pharmacoeconomics. We believe this provides a more comprehensive context for the study.
- The figure summarizing and clarifying the results should be added.
Authors: We appreciate your recommendation. A figure summarizing the results has been added as Figure 2 in the main manuscript to enhance clarity and ease of understanding.
- Following references should be added and properly cited within the main text:
- Turczyn P, Wojdasiewicz P, Poniatowski ŁA, Purrahman D, Maślińska M, Żurek G, Romanowska-Próchnicka K, Żuk B, Kwiatkowska B, Piechowski-Jóźwiak B, Szukiewicz D. Omega-3 fatty acids in the treatment of spinal cord injury: untapped potential for therapeutic intervention? Mol Biol Rep. 2022 Nov;49(11):10797-10809. doi: 10.1007/s11033-022-07762-x.
- Saadoun S, Grassner L, Belci M, Cook J, Knight R, Davies L, Asif H, Visagan R, Gallagher MJ, Thomé C, Hutchinson PJ, Zoumprouli A, Wade J, Farrar N, Papadopoulos MC. Duroplasty for injured cervical spinal cord with uncontrolled swelling: protocol of the DISCUS randomized controlled trial. Trials. 2023 Aug 7;24(1):497. doi: 10.1186/s13063-023-07454-2.
- Kubaszewski Ł, Wojdasiewicz P, Rożek M, Słowińska IE, Romanowska-Próchnicka K, Słowiński R, Poniatowski ŁA, Gasik R. Syndromes with chronic non-bacterial osteomyelitis in the spine. Reumatologia. 2015;53(6):328-36. doi: 10.5114/reum.2015.57639.
- Kimura A, Suehiro K, Mukai A, Fujimoto Y, Funao T, Yamada T, Mori T. Protective effects of hydrogen gas against spinal cord ischemia-reperfusion injury. J Thorac Cardiovasc Surg. 2022 Dec;164(6):e269-e283. doi: 10.1016/j.jtcvs.2021.04.077. Epub 2021 May 4. PMID: 34090694.
Authors: Thank you for suggesting these valuable references. We have incorporated them into the manuscript and cited them appropriately to strengthen the paper's scientific rigor.
- In some places the use of English could be improved on.
Authors: We have carefully reviewed the manuscript and made the necessary language corrections for improved clarity and readability.
Completing this gaps will have an impact on the understanding the aim of the study and, from my point of view, is absolutely necessary.
Comments on the Quality of English Language
minor changes
Authors: We have made all the suggested changes in the manuscript, which can be easily tracked in the Word document using the 'Track Changes' feature. We appreciate your valuable feedback and hope that these revisions meet your approval.
